# MAST: Masked Augmentation Subspace Training for Generalizable Self-Supervised Priors

**Chen Huang, Hanlin Goh, Jiatao Gu & Josh Susskind**
Apple Inc.
`{chen-huang,hanlin,jgu32,jsusskind}@apple.com`

## Abstract

Recent Self-Supervised Learning (SSL) methods are able to learn feature representations that are invariant to different data augmentations, which can then be transferred to downstream tasks of interest. However, different downstream tasks require different invariances for their best performance, so the optimal choice of augmentations for SSL depends on the target task. In this paper, we aim to learn self-supervised features that generalize well across a variety of downstream tasks (*e.g.*, object classification, detection and instance segmentation) without knowing any task information beforehand. We do so by *Masked Augmentation Subspace Training* (or MAST) to encode in the single feature space the priors from different data augmentations in a factorized way. Specifically, we disentangle the feature space into separate subspaces, each induced by a learnable mask that selects relevant feature dimensions to model invariance to a specific augmentation. We show the success of MAST in jointly capturing generalizable priors from different augmentations, using both unique and shared features across the subspaces. We further show that MAST benefits from uncertainty modeling to reweight ambiguous samples from strong augmentations that may cause similarity mismatch in each subspace. Experiments demonstrate that MAST consistently improves generalization on various downstream tasks, while being task-agnostic and efficient during SSL. We also provide interesting insights about how different augmentations are related and how uncertainty reflects learning difficulty.

## 1 Introduction

Self-Supervised Learning (SSL) for image representation has made significant progress over the past few years. The feature representations are typically learned to be invariant to different data augmentations (*e.g.*, *Random Flip* and *Color Jitter*). For example, the popular contrastive SSL methods (Chen et al., 2020a; He et al., 2020) learn invariances by discriminating augmented views of the same image (positive pair) from those of different images (negative pair), while recent non-contrastive SSL methods (Chen & He, 2021; Grill et al., 2020; Bardes et al., 2022) simply maximize the similarity between positive pairs. Such learned features are shown to generalize across many downstream tasks, including classification, object detection, instance segmentation, *etc*.

Despite achieving strong transfer performance, we lack a good theoretical understanding about why both contrastive and non-contrastive SSL methods generalize so well. Balestriero & LeCun (2022) recently proposed a unified framework that demonstrates the key for generalization lies in the alignment between the pairwise relation in SSL (characterized by augmented inputs) and downstream task. This is also in line with other theories (Arora et al., 2019; HaoChen et al., 2021) that quantify how data augmentations implicitly encode the class distributions in downstream tasks. Motivated by these theoretical analyses, we aim at a working SSL method that can directly capture meaningful priors from data augmentations in order to encourage generalization for a range of tasks.

Invariance, as achieved through augmentation, is a useful mechanism to facilitate generalization. For example, one can imagine that invariance to *Random Flip* will boost generalization on many vision tasks. However, as shown in Fig. 1, the optimal set of augmentations (thus invariances) highly depends on the downstream task, which has been similarly observed in (Tian et al., 2020). Sometimes, invariances (*e.g.*, to *Color Jitter*) that prove helpful for one downstream task (*e.g.*, object detection)

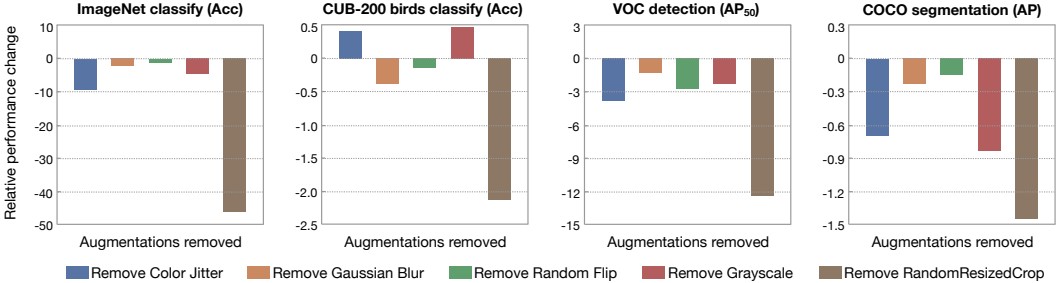

Figure 1: The change of downstream performance per task due to the selective removal of each augmentation during SSL, relative to one baseline SSL method MoCo (He et al., 2020) trained with 5 standard data augmentations, including *Color Jitter, Gaussian Blur, Random Flip, Random Grayscale, RandomResizedCrop*. Performance drop after removing one augmentation indicates that it **helps** the corresponding task, while performance gain indicates **harm**. We observe that *Random-ResizedCrop* strongly benefits all tasks. Other augmentations encode their specific invariances that can be more helpful for some tasks than others. Some augmentations (*e.g.*, *Color Jitter*) even hurt one task (CUB-200 bird classification) despite being quite helpful for other tasks.

may even hurt generalization on another (*e.g.*, birds classification which requires accurate color information of similarly shaped bird species). Hence it is impossible to maximize generalization by finding good augmentations for SSL *in a task-dependent way*, not only because the invariances learned across augmentations may contradict each other, but also because we often do not know the target task *a priori* during SSL. Furthermore, manually finding suitable augmentations for generalizing to a new task is quickly cumbersome.

In this paper, we propose a new SSL method to learn generalizable features without presuming any downstream task information. Our method called *Masked Augmentation Subspace Training* (MAST) achieves this goal by learning a single but disentangled feature space to encode a set of potentially contradicting augmentation invariances. For each augmentation, we learn invariance to it in a specialized feature subspace, which is induced from the full space with a learned masking operation. This allows us to learn unique features for each augmentation subspace as well as features shared between them. There are two main benefits of such subspace training: 1) we explicitly avoid feature suppression (Li et al., 2020) by jointly training separate feature subspaces, such that the learning of one subspace will not compromise that of another; 2) we obtain a disentangled, full feature space that is pre-trained in a task-agnostic way, but does not discard the diverse information (*i.e.*, invariances) required by all possible downstream tasks. We further model uncertainties in the feature representations to reduce harm of ambiguous samples from strong augmentations.

In order to examine how representation effectiveness scales with augmentation diversity, we run experiments with different numbers of augmentations, starting from 5 standard augmentations used typically in SSL (Chen et al., 2020a), and extending to an additional 10 augmentations (totaling 15). Note Tian et al. (2020) and Wang & Qi (2021) also learn from stronger augmentations, but not in our factorized and uncertainty-aware fashion. When it comes to transfer learning on downstream tasks, we simply drop our subspace masks and finetune the full feature representations for high efficiency. This is in contrast to LooC (Xiao et al., 2021) which needs to combine multiple feature "heads", with one head being invariant to all augmentations and other heads being sensitive to a particular augmentation but invariant to others. Both the "leave-one-out" training and feature ensembling strategies in LooC lead to redundancy of parameters and high cost (thus only allowing a few augmentation-specific heads).

Experiments show that MAST, while being efficient and task-agnostic, achieves state-of-the-art transfer performance on diverse downstream vision tasks. Investigations of the subspace masks and uncertainties also provide interesting insights in how different augmentations are related, and in how uncertainty reflects learning difficulty to avoid similarity mismatch during invariance learning.

To summarize, here are our **main contributions**:

- We introduce MAST to make SSL representations disentangled and uncertainty-aware to effectively encode different augmentation invariances for good generalization.
- We show MAST is efficient, is resistant to feature suppression, and achieves state-of-the-art downstream performance on diverse vision tasks without presuming any task information during pre-training.

- We provide interesting insights about how different augmentations are related in SSL and how uncertainty reflects learning difficulty and impacts learning adaptively.

## 2 RELATED WORK

**Self-Supervised Learning.** Contrastive learning is one popular form of SSL, which contrasts different views of an input (positive pairs) against other inputs (negative pairs). Recent contrastive methods mainly differ in how they sample negative pairs, *e.g.*, from a large batch as in SimCLR (Chen et al., 2020a) or from a memory bank as in MoCo (v2) (He et al., 2020; Chen et al., 2020b). One downside of contrastive methods is that they require a multitude of negative pairs to work well, leading to memory inefficiency. Recent non-contrastive SSL methods such as BYOL (Grill et al., 2020), SimSiam (Chen & He, 2021) and OBoW (Gidaris et al., 2021) improve by not using negative pairs. They only maximize the consistency between positive pairs in a teacher-student framework, and some variants like SwAV (Caron et al., 2020), Barlow Twins (Zbontar et al., 2021) and VICReg (Bardes et al., 2022) achieve state-of-the-art performance on several downstream tasks. Note Tian et al. (2020) and Wang & Qi (2021) point out the important role of using strong data augmentations on generalization, but none of the aforementioned works explore the connection between learning factorized augmentation invariances and generalization.

**Theoretical analysis on generalization.** There has been significant interest in theoretically explaining the success of contrastive SSL from various perspectives, *e.g.*, mutual information maximization (Bachman et al., 2019) or dimension collapse (Jing et al., 2022). Most relevant to our work are theoretical studies (Arora et al., 2019; HaoChen et al., 2021) that show the augmentation distributions of same-class inputs have sufficient overlap when the costrastive loss is small, in which case data augmentations can implicitly encode the class structure for downstream tasks. Balestriero & LeCun (2022) further prove that generalization depends on the alignment between pairwise data relations with augmentations and downstream task. Inspired by these theories based on data and augmentation distributions, we propose to improve the generalization of SSL by a factorized learning of augmentation properties as well as uncertainty modeling. The factorization allows each downstream task to choose which underlying symmetries in the data require specific invariances.

**Empirical works on generalization.** Designing better augmentation strategies is one effective way to create task-relevant inductive biases for generalization. Tian et al. (2020) propose to *learn* the augmented views, but require labeled data from the target task, which is not scalable. Another line of works study the feature suppression phenomenon in SSL that can negatively impact downstream performance via "shortcuts" (*i.e.*, by suppressing useful predictive features). To avoid shortcut solutions, Robinson et al. (2021) propose implicit feature modification to encourage instance discrimination using a wider variety of features; while Li et al. (2020) force contrastive learning to learn features that can also predict the input, without discarding important information. Our method automatically avoids shortcuts by jointly training multiple masked feature subspaces within the full space. Similar to our subspace training idea is LooC (Xiao et al., 2021) where both augmentation-variant and -invariant feature spaces are learned and then concatenated to support different tasks. By contrast, our MAST is parameter-efficient by learning a single feature space, can capture a lot more augmentation invariances efficiently in masked subspaces, and is aware of feature uncertainties.

**Uncertainty modeling.** Uncertainty-aware methods (Gal, 2016) have been extensively studied in supervised learning to handle ambiguous or noisy inputs. For SSL, only a few studies estimate uncertainties helpful in specific applications, such as self-supervised depth estimation (Poggi et al., 2020) and multiview stereo (Xu et al., 2021). General methods for uncertainty-aware SSL are lacking in the literature. One exception is variational inference SimSiam (Nakamura et al., 2022), a probabilistic extension of the non-contrastive SSL method SimSiam (Chen & He, 2021) that models uncertainty using spherical posterior distributions. Here we propose a simple alternative for uncertainty modeling in our SSL framework, where a Gaussian feature assumption suffices in each of our augmentation-specific subspaces.

## 3 MAST: MASKED AUGMENTATION SUBSPACE TRAINING

We propose a new SSL method to learn generalizable priors without presuming any downstream task information. To achieve this goal, we identify three desiderata for our method: 1) effective in capturing distinct priors (about invariances) from different augmentations. This will ease transfer learning

Figure 2: **MAST: Masked Augmentation Subspace Training** for SSL. Given an input image $x$, two augmented views $v$ and $v'$ are produced by composing $K$ image augmentations sampled from a distribution $\mathcal{T}$. Then different views are encoded into their feature representations, followed by a projection into Gaussian embeddings $\mathcal{N}(\mu, \Sigma)$ encoding uncertainty. For a disentangled learning of $K$ types of augmentation invariances, we jointly learn a mask $m_k$ for each augmentation to attend to the corresponding subspace where a similarity loss is computed. After pretraining, everything but the encoder $f$ is discarded. The representations of the encoder are used for downstream tasks, without requiring overhead from ensembling subspace features or masks during transfer learning.

on different downstream tasks, where each task automatically selects useful priors in the pre-trained representations via finetuning; 2) free of feature suppression (Li et al., 2020), *i.e.*, learning priors in some features comes at no cost of suppressing others. Particularly, we should accommodate learning different invariances that can be either universally helpful for all downstream tasks or that would otherwise contradict each other (as shown in Fig. 1); 3) efficient enough to learn a large number of priors without too much cost (time or memory), and incurs no overhead for transfer learning.

## 3.1    DESCRIPTION OF MAST

We achieve the desiderata above by MAST, see the schematic in Fig. 2. Assume we have a sampled batch of $n$ images, and an augmentation set $\mathcal{T}$ consisting of $K$ atomic augmentation operators $o_1, \ldots, o_K$ such as *RandomResizedCrop* and *Color Jitter*. For each image $x$, we produce two augmented views $v = t(x)$ and $v' = t'(x)$ by sampling random compositions of image augmentations $t \sim \mathcal{T}_{(o_1, \ldots, o_K)}$ and $t' \sim \mathcal{T}_{(o_1, \ldots, o_K)}$ with their specific augmentation parameters. We encode the views $v$ and $v'$ by an encoder $f$ into their representations $y = f(v)$ and $y' = f(v')$, which are then mapped by a projector $g$ onto embeddings $z = g(y)$ and $z' = g(y')$.

Typically, a similarity loss is computed between $z$ and $z'$ to learn augmentation invariances. However, this makes it hard to learn distinct invariances because learning is unfactorized in the presence of the compositions of $K$ augmentations applied on $x$. Even worse, unfactorized learning from contradicting augmentation compositions could cause feature suppression that hurts generalization.

**Disentangled learning** is the main idea behind our MAST method to learn different invariances without feature suppression. Let $Z = [z_1, \ldots, z_n]$ and $Z' = [z'_1, \ldots, z'_n]$ be the two batches of $d$-dimensional embeddings of the augmented views $V = [v_1, \ldots, v_n]$ and $V' = [v'_1, \ldots, v'_n]$. We build MAST on top of the VICReg framework (Bardes et al., 2022) which minimizes the distance between the embeddings of different views, with two regularization terms:

$$\ell(Z, Z') = \lambda D(Z, Z') + \alpha \ell_{var}(Z, Z') + \beta \ell_{cov}(Z, Z'), \tag{1}$$

where $D(Z, Z')$ enforces augmentation invariance in form of the mean-squared euclidean distance $\sum_i ||z_i - z'_i||_2^2/n$, without any feature normalization. The variance term $\ell_{var}(Z, Z')$ keeps the information content of each embedding (in $Z$ or $Z'$) above a certain level. While the covariance term $\ell_{cov}(Z, Z')$ is introduced to prevent informational collapse by decorrelating different dimensions in each embedding. $\lambda, \alpha$ and $\beta$ are the loss coefficients.

The above variance and covariance terms result in maximum information embeddings that are well suited for disentangled learning. As such, we propose to disentangle the embedding space into distinct subspaces, each of which captures invariance to a specific augmentation. Instead of using bulky MLP heads to generate subspaces, we introduce $K$ masks $M \in \mathbb{R}_{\geq 0}^{d \times K}$ over the embedding space at a much lower cost. Specifically, each mask $m_k = M_{:,k}$ is associated with a fixed augmentation operator $o_k$ from set $\mathcal{T}$, and $m_k$ has the same dimensionality $d$ as the embedding vector $z_i$.

To attend to the $k$th subspace, we only need to apply an element-wise masking operation $\odot$ using the corresponding mask $m_k$ to select relevant embedding dimensions (*i.e.*, $z_i \odot m_k$). Therefore, to learn

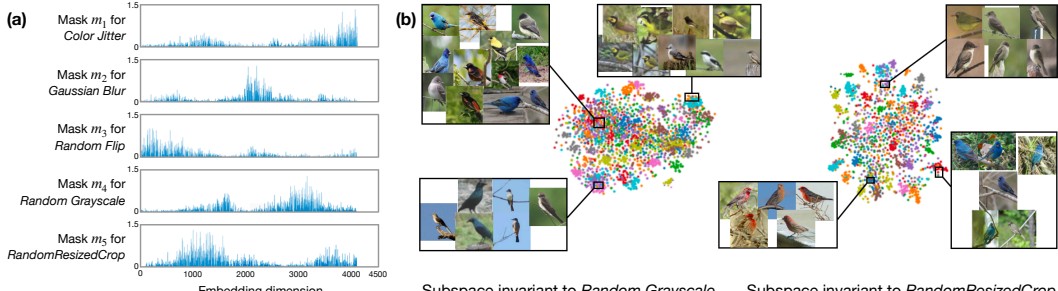

Figure 3: (a) Visualizing the masks learned for 5 augmentation subspaces on ImageNet. The masks disentangle the full embedding space but allow shared dimensions between different subspaces. (b) 2D visualizations of two example subspaces on CUB-200 dataset (Wah et al., 2011). The subspaces are obtained by masking the ImageNet-pretrained feature embeddings using the learned masks. Different colors correspond to different classes. We observe meaningful subspaces with desired invariances: in the subspace learned to be invariant to the *RandomResizedCrop* augmentation, classes are reasonably separated probably because random image patches can exploit color cues that are important for fine-grained birds classification; while the subspace invariant to the *Random Grayscale* augmentation is less discriminative (only for this particular task), because such invariance discards color information so one has to use other cues, *e.g.*, birds' pose and shape for classification.

disentangled augmentation invariances from the paired views $V$ and $V'$, we replace the unfactorized distance function in Eq. (1) with the sum of $K$ masked embedding distances:

$$D_{me}(Z, Z'; M) = \frac{1}{n} \sum_{i=1}^{n} \sum_{k=1}^{K} ||z_i \odot m_k - z_i' \odot m_k||_2^2, \tag{2}$$

$$\ell_{sp}(M) = ||M||_1, \tag{3}$$

where we mask $K$ times for the original embedding $z_i$ of an augmented view with $K$ augmentations. The term $\ell_{sp}(M)$ helps to encourage sparse masks. Note the embeddings and masks are jointly trained (see Section 3.2 for more training details). Fig. 3 shows example masks that induce 5 distinct but overlapping subspaces. This not only avoids potential feature suppression in separate subspaces, but also encourages collaborative learning in the shared embedding dimensions. Fig. 6 (Appendix C) reaffirms that different augmentation subspaces benefit different tasks.

Eventually, we obtain a single but disentangled embedding space that stores different invariance priors for generalization. Note the disentanglement at the embedding level will have a disentanglement effect at the representation level. Thus, we only use the feature representations for testing on downstream tasks, dropping all the pre-trained embeddings and subspace masks. As a result, we introduce no computation overhead from subspace ensembling during transfer learning.

**Uncertainty modeling** plays an important role when the input is ambiguous (*e.g.*, due to strong augmentations), but still remains a largely unexplored problem in SSL. We address this issue with a simple method that explicitly models uncertainty using Gaussian embeddings. Concretely, instead of having a projector $g : y \to z$ to map representations $y$ to deterministic embeddings $z \in \mathbb{R}^d$, we extend to the projector $g : y \to (\mu, \Sigma)$ that produces a multivariate Gaussian distribution $\mathcal{N}(\mu, \Sigma)$ with mean $\mu \in \mathbb{R}^d$ and covariance $\Sigma \in \mathbb{R}^{d \times d}$. We consider $\Sigma$ to be diagonal for efficiency, modeling the uncertainty independently for every dimension of the mean embedding $\mu$.

With this independence assumption, we can use masking to easily induce embedding subspaces that are still Gaussian, *i.e.*, $\mathcal{N}(\tilde{\mu}_k, \tilde{\Sigma}_k)$ where $\tilde{\mu}_k = \mu \odot m_k$ and $\tilde{\Sigma}_k = \Sigma \odot \text{diag}(m_k)$. We find the Gaussian embeddings suffice to model the uncertainties of augmented views in each subspace where only one type of augmentation is involved. Then we further improve Eq. (2) to learn factorized augmentation invariances in an uncertainty-aware manner:

$$D_{mg}(V, V'; M) = \frac{1}{n} \sum_{i=1}^{n} \sum_{k=1}^{K} \frac{2||\tilde{\mu}_k^i - \tilde{\mu}_k'^i||_2^2}{\text{tr}(\tilde{\Sigma}_k^i) + \text{tr}(\tilde{\Sigma}_k'^i)}, \tag{4}$$

where $\tilde{\mu}_k^i$ and $\tilde{\mu}_k'^i$ are the mean embeddings of views $v_i$ and $v_i'$ in the $k$th subspace, and their distance is re-weighted by their uncertainty measures (trace $\text{tr}(\cdot)$ of covariance matrix) to determine

the closeness of the two Gaussians. Intuitively, when the embeddings have high uncertainties, the similarity loss should be downweighted to prevent similarity mismatch between ambiguous views.

To stablize the learning of uncertainties, we introduce an extra regularization term to encourage consistency between the predicted Gaussian embeddings $\mathcal{N}_i(\mu_i, \Sigma_i)$ and $\mathcal{N}'_i(\mu'_i, \Sigma'_i)$ for views $v_i$ and $v'_i$. This term is based on symmetric KL divergence, with a closed-form expression for Gaussians:

$$\ell_{kl}(V, V') \; = \frac{1}{n} \sum_{i=1}^{n} D_{KL}(\mathcal{N}_i || \mathcal{N}'_i) + D_{KL}(\mathcal{N}'_i || \mathcal{N}_i), \tag{5}$$

$$D_{KL}(\mathcal{N}_i || \mathcal{N}'_i) \; = \frac{1}{2} \left[ \text{tr} \left( \Sigma_i'^{-1} \Sigma_i \right) + (\mu'_i - \mu_i)^T \Sigma_i'^{-1} (\mu'_i - \mu_i) + \log \frac{|\Sigma'_i|}{|\Sigma_i|} \right]. \tag{6}$$

Combining Eqs. (3-6) and substituting them into Eq. (1), we arrive at our final loss function:

$$\ell(V, V'; M) = \lambda D_{mg}(V, V'; M) + \lambda_1 \ell_{sp}(M) + \lambda_2 \ell_{kl}(V, V') + \alpha \ell_{var}(V, V') + \beta \ell_{cov}(V, V'). \tag{7}$$

## 3.2 IMPLEMENTATION DETAILS

**Architecture and hyper-parameters.** Our pretraining is mostly performed on the unlabeled ImageNet dataset (Deng et al., 2009) with 1000 classes unless otherwise stated. For fair comparison, the encoder $f$ is the widely-adopted ResNet-50 backbone (He et al., 2016) which outputs 2048-dimensional feature representations. The projector $g$ has two fully-connected layers of size 8192, followed by separate fully-connected heads that respectively output the mean $\mu$ and diagonal covariance $\Sigma$ (both of $d = 4096$ dimensions) of Gaussian embeddings. Loss coefficients in Eq. (7) are set as $\alpha = 25$, $\beta = 1$ following VICReg (Bardes et al., 2022), and we set $\lambda = 25d/K$, $\lambda_1 = 600/(dK)$ and $\lambda_2 = 25$ to generate comparable loss magnitudes. Fig. 8 in Appendix D shows the pretraining performance on different datasets is not very sensitive to the loss coefficients.

**Data augmentations.** Our MAST-5 method uses $K = 5$ augmentations adopted by most recent SSL methods for fair comparison purposes. The augmentations include *Color Jitter (of brightness, contrast, saturation, hue), Gaussian Blur, Random Flip, Random Grayscale, RandomResizedCrop*. While MAST-15 uses 10 more augmentations (hence $K = 15$). The added augmentations are *ShearX(Y), TranslateX(Y), Rotate, Invert* borrowed from RandAugment Cubuk et al. (2019), as well as *Sharpness, Gaussian Noise, Sobel Filter* and *Cutout* (DeVries & Taylor, 2017) for diversity.

**Training details**: The training protocol follows that in (Bardes et al., 2022): with batch size 2048, the LARS optimizer (You et al., 2017) runs for 1000 epochs with weight decay $10^{-6}$ and learning rate 1.6. The learning rate follows a cosine annealing schedule (Loshchilov & Hutter, 2017). For effective learning of disentangled subspaces, we found it better to schedule the available augmentations in two stages. In the first 500 epochs, we sample one random augmentation $o_k$ at a time to train the corresponding subspace and mask $m_k$, setting $K = 1$ in Eq. (4). This ensures all the factorized subspaces are trained well in this stage. For the remaining 500 epochs, random augmentation compositions $(\dots, o_k, \dots)$ are allowed to learn disentangled invariances from progressively stronger augmentations ($K$ linearly increases from 1 to its maximum value). For each composing augmentation, we use its default augmentation probability but random magnitude for robust training.

## 4 EXPERIMENTS

**Main results.** We start with evaluating our MAST method for SSL on ImageNet. Table 1 shows the results from both linear and semi-supervised evaluations, using the same optimization procedures of our baseline VICReg (Bardes et al., 2022). The numbers reported for our MAST-5 and MAST-15 variants are the mean scores from 3 runs with different random initialization.

It can be seen from Table 1 that when 5 standard augmentations are used for SSL, MAST-5 achieves strong performance on-par with or better than that of state-of-the-art contrastive and non-contrastive methods. MAST-5 also consistently outperforms our baseline method VICReg, indicating the power of disentangled and uncertainty-aware learning of different augmentation invariances. Note predictive contrastive learning (PCL) is related to us in avoiding feature suppression, but by strengthening contrastive learning with a predictive inpainting task. MAST-5, on the other hand, achieves much

Table 1: **Evaluation on ImageNet** using representations trained with a ResNet-50 backbone in terms of Top-1 and Top-5 accuracies (%). We evaluate linear classification using the frozen representations from ImageNet, and semi-supervised classification using the fine-tuned representations from 1% and 10% ImageNet samples. Most methods in the middle cell (including our MAST-5) use the same 5 augmentations in (Chen et al., 2020a), while the bottom cell compares methods using much more augmentations, *e.g.*, from RandAugment (Cubuk et al., 2019).

| Method | Linear | | Semi-supervised | | | |
|---|---|---|---|---|---|---|
| | Top-1 | Top-5 | Top-1 | | Top-5 | |
| | | | 1% | 10% | 1% | 10% |
| Supervised | 76.5 | - | 25.4 | 56.4 | 48.4 | 80.4 |
| MoCo (He et al., 2020) | 60.6 | - | - | - | - | - |
| SimCLR (Chen et al., 2020a) | 69.3 | 89.0 | 48.3 | 65.6 | 75.5 | 87.8 |
| PCL (Li et al., 2020) | 71.0 | - | - | - | - | - |
| MoCo v2 (Chen et al., 2020b) | 71.1 | - | - | - | - | - |
| SimSiam (Chen & He, 2021) | 71.3 | - | - | - | - | - |
| SwAV (Caron et al., 2020) | 71.8 | - | - | - | - | - |
| OBoW (Gidaris et al., 2021) | 73.8 | - | - | - | **82.9** | **90.7** |
| BYOL (Grill et al., 2020) | 74.3 | 91.6 | 53.2 | 68.8 | 78.4 | 89.0 |
| SwAV (w/ multi-crop) (Caron et al., 2020) | **75.3** | - | 53.9 | **70.2** | 78.5 | 89.9 |
| Barlow Twins (Zbontar et al., 2021) | 73.2 | 91.0 | 55.0 | 69.7 | 79.2 | 89.3 |
| VICReg (Bardes et al., 2022) | 73.2 | 91.1 | 54.8 | 69.5 | 79.4 | 89.5 |
| **MAST-5 (ours)** | 74.9 | **91.8** | **55.2** | **70.2** | 80.1 | 90.2 |
| InfoMin Aug (Tian et al., 2020) | 73.0 | 91.1 | - | - | - | - |
| CLSA (Wang & Qi, 2021) | 72.2 | - | - | - | - | - |
| **MAST-15 (ours)** | **75.8** | **92.1** | **55.8** | **71.4** | **81.0** | **90.9** |

Table 2: **Transfer performance on downstream tasks** using the pretrained representations on ImageNet. We evaluate linear classification using frozen representations on datasets Places205 (Top-1 accuracy %), VOC07 (mAP %) and iNat18 (Top-1 accuracy %). We also evaluate fine-tuning performance for object detection on VOC07+12 ($AP_{50}$) and for instance segmentation on COCO (AP).

| Method | Linear Classification | | | Detection | Segmentation | |
|---|---|---|---|---|---|---|
| | Places205 | VOC07 | iNat18 | VOC07+12 | COCO det | COCO seg |
| Supervised | 53.2 | 87.5 | 46.7 | 81.3 | 39.0 | 35.4 |
| MoCo (He et al., 2020) | 46.9 | 79.8 | 31.5 | - | - | - |
| SimCLR (Chen et al., 2020a) | 52.5 | 85.5 | 37.2 | - | - | - |
| MoCo v2 (Chen et al., 2020b) | 51.8 | 86.4 | 38.6 | 82.5 | 39.8 | 36.1 |
| SimSiam (Chen & He, 2021) | - | - | - | 82.4 | - | - |
| BYOL (Grill et al., 2020) | 54.0 | 86.6 | 47.6 | - | 40.4 | 37.0 |
| SwAV (w/ multi-crop) (Caron et al., 2020) | 56.7 | 88.9 | 48.6 | 82.6 | 41.6 | **37.8** |
| OBoW (Gidaris et al., 2021) | 56.8 | 89.3 | - | 82.9 | - | - |
| Barlow Twins (Zbontar et al., 2021) | 54.1 | 86.2 | 46.5 | 82.6 | 40.0 | 36.7 |
| VICReg (Bardes et al., 2022) | 54.3 | 86.6 | 47.0 | 82.4 | 39.4 | 36.4 |
| **MAST-5 (ours)** | **58.9** | **91.1** | **49.5** | **83.1** | **41.9** | 37.2 |
| InfoMin Aug (Tian et al., 2020) | - | - | - | 82.7 | 40.6 | 36.7 |
| CLSA (Wang & Qi, 2021) | - | **93.6** | - | 83.2 | 42.3 | - |
| **MAST-15 (ours)** | **59.6** | 92.8 | **50.2** | **84.0** | **43.1** | **38.3** |

stronger performance by *explicitly* training different feature subspaces that can encode contradicting invariances without undesired trade-offs. Fig. 3 supports this hypothesis with one visual example of bird classification where one subspace is sensitive to color cues, and the other captures invariance to color, exploiting pose and shape instead. Our method can be further extended to MAST-15 learning with 15 types of augmentations, which outperforms InfoMin Aug and CLSA that similarly learn from stronger augmentation compositions but in a suboptimal unfactorized way.

Next, following the setup in (Bardes et al., 2022), we evaluate the transfer performance of the ImageNet-pretrained representations on various downstream tasks: linear classification on the scene dataset Places205 (Zhou et al., 2014), multi-label image classification dataset VOC07 (Everingham et al., 2010) and fine-grained image classification dataset iNaturalist2018 (Van Horn et al., 2018).

Table 3: **Evaluation of linear classification on ImageNet-100** using representations pre-trained with a ResNet-50 backbone on the same dataset in terms of Top-1 and Top-5 accuracies (%). Ensemble: feature ensembling. Repr: effective representation dimensionality used for classification.

| Method | Top-1 | Top-5 | Repr. |
|---|---|---|---|
| MoCo v2 (Chen et al., 2020b) | 81.0 | 95.2 | 2048 |
| IFM-MoCo v2 (Robinson et al., 2021) | 81.4 | - | 2048 |
| LooC (Xiao et al., 2021) | 79.2 | 94.7 | 2048 |
| LooC++ (Xiao et al., 2021) (ensemble) | 81.2 | 95.2 | 2048×3 |
| MAST-5 (default) | 81.9 | 95.4 | 2048 |
| MAST-5 (ensemble) | 82.1 | 95.4 | 2048×5 |
| MAST-15 | **82.5** | **95.5** | 2048 |

Table 4: **Downstream classification accuracies** (%) of ImageNet-100 pretrained representations on CUB-200 and Flowers-102. $*$ indicates enhanced baselines by AugSelf authors. We compare accuracy gains $_{in\ subscripts}$ over each baseline.

| Method | CUB-200 | | Flowers-102 | |
|---|---|---|---|---|
| | Top-1 | Top-5 | 5-shot | 10-shot |
| MoCo v2 (Chen et al., 2020b) | 36.7 | 64.7 | 67.9 | 77.3 |
| LooC (Xiao et al., 2021) | 39.6 $_{2.9}$ | 69.2 $_{4.5}$ | 70.9 $_{3.0}$ | 80.8 $_{3.5}$ |
| MoCo v2* (Chen et al., 2020b) | 32.2 | - | 78.5 | 81.2 |
| + AugSelf (Lee et al., 2021) | 37.0 $_{4.8}$ | - | 81.7 $_{3.2}$ | 84.5 $_{3.3}$ |
| SimSiam* (Chen & He, 2021) | 38.4 | - | 83.6 | 85.9 |
| + AugSelf (Lee et al., 2021) | 45.3 $_{6.9}$ | - | 86.4 $_{2.8}$ | 88.3 $_{2.4}$ |
| VICReg (Bardes et al., 2022) | 37.5 | 65.2 | 68.3 | 77.5 |
| MAST-5 | 40.3 $_{2.8}$ | 69.8 $_{4.6}$ | 70.1 $_{1.8}$ | 81.2 $_{3.7}$ |
| MAST-15 | 41.1 $_{3.6}$ | 70.0 $_{4.8}$ | 72.4 $_{\mathbf{4.1}}$ | 81.9 $_{\mathbf{4.4}}$ |

We also transfer the representations to object detection on VOC07+12 using Faster R-CNN (Ren et al., 2015) with a R50-C4 backbone, and instance segmentation on COCO (Lin et al., 2014) using Mask-R-CNN (He et al., 2017) with a R50-FPN backbone. Table 2 shows larger performance gap between MAST-5/15 and prior works for almost all tasks, demonstrating the superior generalization capabilities of our learned invariance priors despite being task-agnostic.

**Comparing with more related methods:** Implicit feature modification (IFM) (Robinson et al., 2021) and Leave-one-out Contrastive Learning (LooC) (Xiao et al., 2021). All comparing methods use the same 100-class ImageNet dataset and 500 epochs for pretraining. IFM is a method that avoids shortcut solutions by altering sample features during contrastive learning. Table 3 shows IFM improves the MoCo v2 baseline by a small margin, but not as much as the gain offered by our disentangled method (MAST-5) that explicitly reduces feature suppression.

LooC combines augmentation-invariant and -variant feature subspaces to reduce the harm of particular invariances for some downstream tasks. Table 3 shows that LooC, when regularized with such subspace training, actually hurts performance on the pretraining dataset ImageNet-100 (in comparison to baseline MoCo v2). This suggests that the "disentanglement" learned by LooC may still suppress certain useful features. By contrast, our disentangled learning with MAST-5 seems more effective, giving notable gains over MoCo v2. Note the advanced LooC++ variant can recover the loss of LooC by ensembling feature subspaces. However, LooC++ cannot generate solid gains over the MoCo v2 baseline, and at the same time suffers from a much larger cost from feature ensembling. When we apply the idea of ensembling (masked) feature subspaces to MAST, we observe slightly improved performance at the cost of increased feature dimensionality. Hence our default MAST-5 approach just uses the original unmasked representations without any ensembling, which is efficient and performant already. MAST-15 further improves performance by learning from more augmentations. Table 4 validates that MAST-5/15 remain competitive with LooC and a recent strong method AugSelf (Lee et al., 2021) on two downstream tasks: fine-grained classification on CUB-200 dataset (Wah et al., 2011), and few-shot classification on Flowers-102 dataset (Nilsback & Zisserman, 2008). Table 9 in Appendix D provides more fine-grained classification results.

**Ablation study** is conducted in Table 5 to evaluate different components of MAST. We use the setup where representations are pretrained on the ImageNet-1k dataset and tested on 4 representative tasks.

Recall that MAST-15 gains a lot over MAST-5 by using more augmentations. We start with asking: "is it more about disentangled learning or just richer augmentations?" Row 2 answers this question by applying the same $K = 15$ augmentations to our baseline VICReg. We can see that this variant falls far behind MAST-15 (row 1), and even behind the baseline (row 4) on classification tasks. This indicates that naively composing more augmentations could hurt downstream performance without disentangled learning to avoid feature suppression. Then for MAST-15 with such capability, will even more augmentations help? We experimented with adding 4 common augmentations *Solarize, Equalize, Posterize, Motion Blur* in row 3 (so $K = 19$). No large improvements are observed probably because the added augmentations are redundant (more on augmentation relations later). We leave for future work the investigation of advanced or *learned* augmentation techniques.

Next, we ablate other components of our method based on MAST-5 for efficiency. Recall that MAST-5 benefits from a two-staged scheduling of augmentations (from one at a time to linearly increasing number of augmentations). We find such augmentation scheduling can improve VICReg as well (row 6), but not enough to compete with MAST-5 (row 5). Lastly, by comparing row 5 to

Table 5: **Ablation study** of different MAST components (ImageNet pretraining → 4 testing tasks).

| Comment | Row | Method | Linear classify | Linear classify | Detection | Segmentation |
| --- | --- | --- | --- | --- | --- | --- |
| | | | ImageNet (Top-1) | Places205 (Top-1) | VOC07+12 (AP$_{50}$) | COCO (AP) |
| Disentangled learning or richer aug.? | 1 | MAST-15 | 75.8 | 59.6 | 84.0 | 38.3 |
| | 2 | VICReg ($K = 15$) | 73.1 | 53.5 | 82.6 | 36.9 |
| Will more aug. help? | 3 | MAST-19 | 75.8 | 59.4 | 84.0 | 38.4 |
| Baseline | 4 | VICReg ($K = 5$) | 73.2 | 54.3 | 82.4 | 36.4 |
| | 5 | MAST-5 | 74.9 | 58.9 | 83.1 | 37.2 |
| Staged aug. schedule | 6 | VICReg ($K = 5$) | 73.4 | 54.6 | 82.4 | 36.5 |
| Handcrafted masks | 7 | MAST-5 | 74.3 | 58.1 | 82.5 | 36.9 |
| No uncertainty | 8 | MAST-5 | 74.1 | 57.5 | 82.7 | 36.8 |

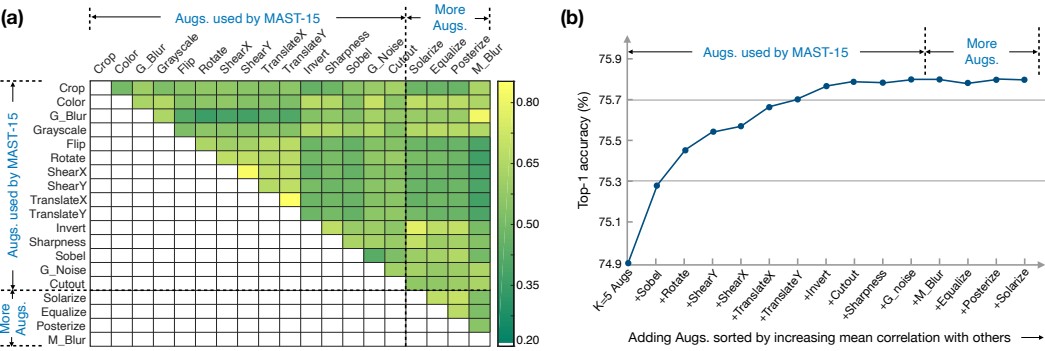

Figure 4: (a) Correlation between augmentations measured by their masks' cosine similarity. (b) ImageNet linear classification Top-1 accuracy vs. increasing number of augmentations during SSL. Augmentations with low correlation with others provide diverse priors and large performance gains.

rows 7-8, we can easily find the benefits of our learned subspace masks (vs. handcrafted disjoint masks: all 1's for each subspace of $d/K$ dimensions, and 0 elsewhere) and uncertainty modeling.

**Visualizations.** We exploit the learned subspace masks to quantify augmentation correlations unsupervisedly by computing the cosine similarity between masks. Fig. 4(a) shows the example pairwise correlation between 19 augmentations on ImageNet. We can observe weakly correlated augmentations such as (*Gaussian Blur*, *Rotate*) and (*Gaussian Blur*, *ShearY*). They define diverse augmentations that could introduce complementary invariance priors for good generalization (if potential contradictions can be reduced, *e.g.*, by our factorized learning). There are also strongly correlated augmentations like (*ShearX*, *ShearY*) that offer redundant invariance information. The 4 optional augmentations *Solarize, Equalize, Posterize, Motion Blur* are highly correlated with others too (*e.g.*, with *Color Jitter* and *Gaussian Blur*), therefore we exclude them from MAST-15. Fig. 4(b) demonstrates the positive role of augmentation diversity on generalization. We gradually add augmentations to MAST-5 for SSL, such that distinct augmentations with low mean correlation with others are added first. Those augmentations are observed to improve the downstream performance quickly, and then performance plateaus with redundant, correlated augmentations. This suggests that our correlation measure can indeed be a rough guidance on composing generalizable augmentations.

Appendix B further shows how MSAT correctly models uncertainties for strongly augmented samples in order to improve SSL performance (comparison with a related uncertainty model (Nakamura et al., 2022) is also provided), and how uncertainty reflects learning difficulty.

## 5 CONCLUSION

This paper studies the important role of data augmentation in creating useful invariance priors during SSL, which is the key inductive bias for good generalization on downstream tasks. Instead of enumerating all the possible augmentation compositions and selecting the best set of invariances for a target task, we propose a unified framework to embed all the distinct priors of invariance into feature representations that can be readily used for different tasks. Our MAST method, featuring disentangled representation learning and uncertainty modeling, proves competitive when evaluated on a variety of downstream vision tasks.

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

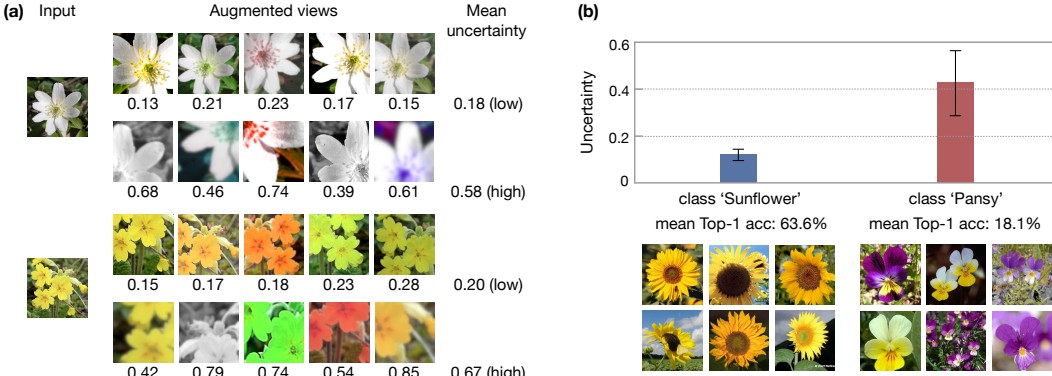

Figure 5: (a) Example uncertainty estimations for augmented views of images on Flowers-17 dataset. Views with strong augmentations have high uncertainties in their feature representations. (b) Mean uncertainty estimations for two flower classes vs. mean linear classification accuracies (Top-1 %) of the two classes. The lower-performing class 'Pansy' has higher uncertainty and larger variance in uncertainty estimates, due to large variations in the flower color, size and shape. This indicates our uncertainty measure can reflect the learning difficulty of the classification task.

## A  ADDITIONAL IMPLEMENTATION DETAILS

**Uncertainty modeling.** We have separate heads in the projector $g$ to predict the mean and diagonal covariance of Gaussian embeddings. The mean head is composed of a Generalized-Mean (GeM) pooling layer (Radenovi et al., 2019) followed by a fully connected layer that outputs $\mu \in \mathbb{R}^d$ with $d = 4096$. The covariance head is composed of a GeM layer and a fully connected layer with ReLU activation function (to ensure positivity) to output the $d$-dimensional diagonal of a covariance matrix $\Sigma$. The separate GeM layers in the mean and covariance heads are beneficial, since they help to learn different $p$-norms for the mean and covariance.

**Mask learning.** Our learned subspace masks act as element-wise gating functions to select relevant embedding dimensions. We ensure the positivity of masks $M \in \mathbb{R}_{\geq 0}^{d \times K}$ by learning them as $M = \max(0, U)$ using a ReLU activation function where $U \in \mathbb{R}^{d \times K}$. During training, we found it useful to initialize the masks as random rather than fixed ones (*e.g.*, when initial mask values are all the same). More specifically, we initialize each mask (*i.e.*, each column of $U$) such that the corresponding subspace of $d/K$ dimensions follows a Gaussian distribution and Gaussian noise is added for all dimensions. The Gaussian prior for each subspace mask eases mask training in the beginning and expedites the search of disentangled solutions. While the added Gaussian noise is large enough (with mean 0.2 and variance 0.01) to help find meaningful mask patterns in all dimensions.

## B  UNCERTAINTY VISUALIZATION AND COMPARISON

**Uncertainty visualization.** In MAST, uncertainty is estimated as a vector, *i.e.*, diagonal covariance of Gaussian embeddings to encode uncertainty in every dimension of the embeddings. For the ease of visualization, we need a scalar uncertainty measure to quantify how uncertain we are about the entire image input and its representations. Here we compute the scalar uncertainty based on a volumetric measure (trace of covariance matrix), which is then linearly rescaled to the range of $[0, 1]$. The higher the value is, the larger uncertainty the representations have. Fig. 5(a) shows some example uncertainty estimations on Flowers-17 dataset (Nilsback & Zisserman, 2006). We observe high uncertainties estimated for strongly augmented images, while low uncertainties for the weakly augmented. This shows our uncertainty measure can reasonably capture the strength of data augmentations. This is a desirable behavior since when the augmentations are strong enough to produce ambiguous views or even change the identity of input images, we can use the estimated high uncertainty to downweight the distance between ambiguous views. As a result, similarity mismatch is avoided during invariance learning.

Table 6: **Evaluation of linear classification on Flowers-17 and Cars** using representations pre-trained on the respective datasets in terms of Top-1 and Top-5 accuracies (%).

| Method | Flowers-17 | | Cars | |
|---|---|---|---|---|
| | Top-1 | Top-5 | Top-1 | Top-5 |
| SimSiam (Chen & He, 2021) | 24.6 | 51.5 | 12.7 | 33.2 |
| VI-SimSiam (Nakamura et al., 2022) | 19.7 | 43.9 | 14.9 | 37.2 |
| MAST-5 (uncertainty only) | 26.5 | 55.7 | 16.4 | 40.3 |
| MAST-5 (uncertainty + disentangled learning) | **28.9** | **59.1** | **18.7** | **44.1** |

Fig. 5(b) further connects uncertainty with the classification performance (linear probing). It can be seen that the 'Pansy' class exhibits high uncertainty and large variance in uncertainty estimates due to the large variations of flower appearances. Accordingly, the 'Pansy' class has a much lower classification accuracy than that of 'Sunflower' class with low uncertainty. This validates that our uncertainty measure can indeed reflect the learning difficulty. For the difficult 'Pansy' class, the large uncertainty variance actually offers the chance for us to adapt invariance learning and avoid similarity mismatch for improvement.

**Comparison.** We compare with VI-SimSiam (Nakamura et al., 2022) that models uncertainty for SSL using spherical posterior distributions and variational inference. We follow VI-SimSiam to pre-train for 100 epochs and use ResNet-18 as the encoder. Pre-training is performed on two datasets respectively: Flowers-17 (Nilsback & Zisserman, 2006) with 17 flower categories (80 images for each category), and Cars (Krause et al., 2013) with 197 car models (16,185 images). After pre-training, we train a linear classifier on the frozen representations using the training set of the corresponding dataset, and evaluate linear classification accuracy in its test set.

Table 6 summarizes the results in Top-1 and Top-5 accuracies. We observe that the uncertainty-aware VI-SimSiam method is not always helpful when compared to its baseline SimSiam (see the notable performance drop on Flowers-17). For MAST-5, we experiment with a special variant 'uncertainty only' to not benefit from disentangled subspace learning (*i.e.*, no masks involved). Note MAST-5 (uncertainty only) is not directly comparable to VI-SimSiam since we have different baselines (VICReg vs. SimSiam) and implementation details. The goal here is only to test how our uncertainty modeling mechanism works in general, when using a simple Gaussian assumption about feature embeddings. Results show that MAST-5 (uncertainty only) obtains quite competitive performance on both datasets, and is further improved by disentangled learning consistently.

## C  MORE ANALYSIS OF THE AUGMENTATION SUBSPACES

**Optimal augmentations (subspaces) are task-dependent.** Fig. 1 supports the claim by demonstrating the difference between the optimal augmentations for different downstream tasks. This motivates us to develop MAST, by learning disentangled augmentation invariances in masked feature subspaces that are augmentation-specific. Now with MAST, we find it much easier to quantify the relationship between augmentations and tasks, using the learned augmentation-specific masks as a bridge. The end task can freely select from the factorized latent space which feature masks or augmentations are relevant to the task. This can help us to identify optimal augmentations for one task in a cheaper way than the leave-one-out strategy in Fig. 1.

We take the classification task as one example. For correctly classified samples, we examine the class predictions (only for the ground-truth class, using the corresponding weights of the classifier) by differently masked feature embeddings $\tilde{\mu}_k^i$ of each sample. The higher the class prediction, the more the corresponding masked augmentation subspace (and its captured invariance) benefits the given task. Fig. 6 shows the class predictions across 5 augmentation subspaces for ImageNet and CUB-200 classification, respectively. We observe that, again, different tasks benefit from a different set of augmentations (*e.g.*, *Color Jitter, Random Grayscale, RandomResizedCrop* for ImageNet, and *Gaussian Blur, Random Flip, RandomResizedCrop* for CUB-200).

**Augmentation invariance vs. preserving augmentation-specific information.** Since our MAST method focuses on disentangled learning of different augmentation invariances, we now verify how

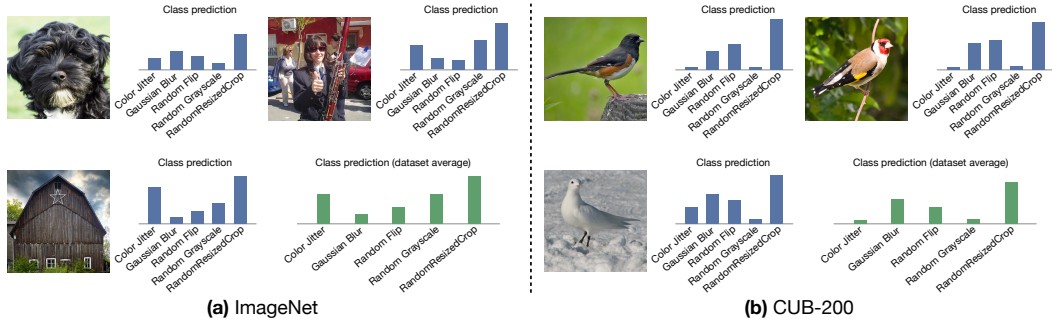

Figure 6: Class predictions by different augmentation-specific subspaces of correctly classified images on (a) ImageNet and (b) CUB-200 datasets. MAST-5 is used here. Note we attend to different embedding subspaces via masking based on the learned masks for 5 augmentations. Class predictions are obtained by passing the masked embeddings through the classifier weights associated with the ground-truth label. The higher the class prediction, the more beneficial the corresponding augmentation subspace (and its captured invariance) is to the classification task. As can be observed, the optimal set of augmentation invariances are different for the two classification tasks.

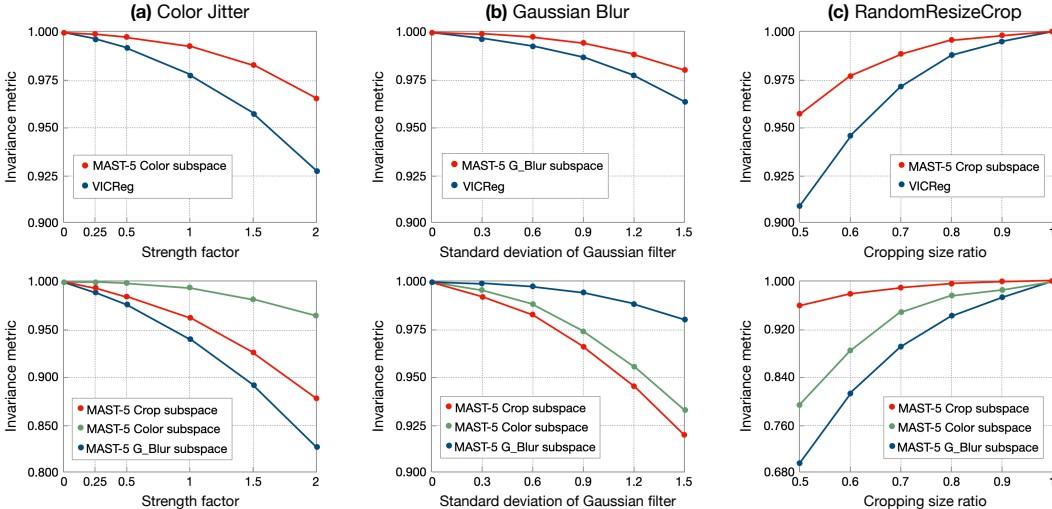

Figure 7: **Invariance metric for ImageNet-pretrained feature embeddings**. The invariance metric is defined as the cosine similarity between masked embeddings from augmented and original samples. The higher the metric, the more invariant the subspace is to the tested augmentation. The invariance metric is computed against 3 augmentations with continuous augmentation parameters: (a) *Color Jitter* with the strength factor multiplied over the default configuration (larger value indicates stronger augmentation), (b) *Gaussian Blur* with the standard deviation of Gaussian filter (larger value indicates stronger augmentation), (c) *RandomResizedCrop* with the cropping size ratio (smaller value indicates stronger augmentation). **Top row:** augmentation invariances are well captured by our masked subspaces (better than baseline VICReg features). **Bottom row:** our subspaces are disentangled since each is specialized to the corresponding augmentation.

well the invariance property is captured in each of our augmentation subspaces. The invariance metric is used by computing the cosine similarity between masked embeddings $\tilde{\mu}_k^i$ from augmented and original samples. Higher metric indicates higher level of augmentation invariance. Fig. 7 shows this metric on ImageNet when we vary the augmentation parameters of 3 standard augmentations – *Color Jitter, Gaussian Blur, RandomResizedCrop*. We exclude other 2 (*Random Flip, Random Grayscale*) whose augmentation parameters are discrete.

Two observations from the figure: 1) in top row, our subspaces show great robustness under various augmentation intensities, especially when compared to the VICReg baseline that learns all augmen-

Table 7: **Linear classification accuracy (%) of rotation** using STL10-pretrained representations.

| Method | SimSiam (Chen & He, 2021) | SimSiam+AugSelf (Lee et al., 2021) | MAST-5 |
|---|---|---|---|
| Accuracy | 59.11 | **64.61** | 63.47 |

Table 8: **ImageNet performance as a function of pretraining epochs.** Performance is measured as the Top-1 accuracy (%) of linear classification. Our default number of pretraining epochs is 1000.

| Epochs | 200 | 400 | 800 | 1000 |
|---|---|---|---|---|
| VICReg (Bardes et al., 2022) | 70.2 | 72.3 | 73.0 | 73.2 |
| MAST-5 | 70.9 | 73.5 | 74.5 | 74.9 |
| $\Delta$ | 0.7 | 1.2 | 1.5 | 1.7 |

tation invariances in an unfactorized way. 2) the bottom row shows the disentanglement of different subspaces, since each subspace demonstrates specialization to the corresponding augmentation.

One might ask: will we loose all augmentation-sensitive information when we learn factorized invariances from those augmentations? Short answer is no, as evidenced by our strong performance on various downstream tasks that rely on specific augmentation characteristics (*e.g.*, color-sensitive bird classification on CUB-200). We hypothesize that different subspaces may offer complementary information about augmentations for different tasks to choose from. For example, in Fig. 3(b), the color information suppressed in the subspace invariant to *Random Grayscale* is available in the subspace invariant to *RandomResizedCrop*.

To further support the above hypothesis, we follow the AugSelf work (Lee et al., 2021) to solve a pretext task of 4-way rotation classification $(0°, 90°, 180°, 270°)$ using pretrained representations. The same pretraining protocol is followed: pretraining ResNet-18 for 200 epochs on STL-10 dataset (Coates et al., 2011), and standard augmentations are used (no *Rotate*). To evaluate rotation classification accuracy, a linear classifier is trained on top of the frozen representations. Table 7 shows competitive performance of MAST-5 (63.47%) vs. AugSelf (64.61%) which is explicitly trained to be augmentation-predictive. This validates that we can learn invariances to standard augmentations while maintaining sensitivity to the unseen rotation. In other words, MAST is able to preserve augmentation-invariant and -specific information implicitly in separate subspaces.

# D    ADDITIONAL RESULTS

**Impact of pretraining epochs.** Recall our MAST models are pre-trained in two stages, with increasing number of augmentations (i.e. stronger compositions). As a result, MAST typically benefits significantly from longer training to take full advantage of disentangled learning with strong augmentation compositions. Table 8 exemplifies the ImageNet performance as a function of pretraining epochs, from 200 to default 1000. We can observe that 1) our MAST-5 outperforms its baseline model VICReg across all epoch numbers, 2) more epochs lead to larger gains, but the gain is still decent (0.7%) at low epoch number 200.

**Sensitivity of MAST to loss coefficients.** Recall the loss function of MAST in Eq. (7) consists of 2 terms borrowed from the VICReg baseline with their default loss coefficients. We introduce 3 more loss terms, and only set their loss coefficients $\lambda = 25d/K$, $\lambda_1 = 600/(dK)$ and $\lambda_2 = 25$ to have a comparable loss magnitude with that of other terms. In practice, we use the exact same loss coefficients across different augmentation distributions and different pretraining datasets. Fig. 8 shows the pretraining performance is not very sensitive to $\lambda, \lambda_1, \lambda_2$, especially when they are scaled by a factor between 0.5 and 2. Note for comprehensive evaluations, we monitor the pretraining performance on the large dataset ImageNet and two relatively small datasets Flowers-17 and Cars, using their representative evaluation tasks. We could have easily tuned $\lambda, \lambda_1, \lambda_2$ per dataset for best performance, but we choose to use the same loss coefficients for simplicity.

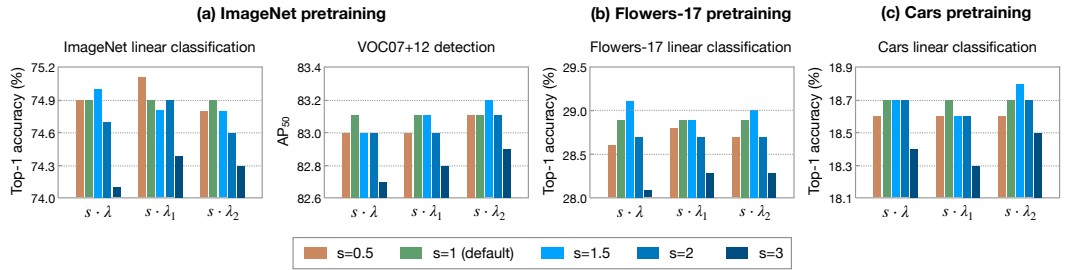

Figure 8: **Sensitivity of our MAST-5 to the loss coefficients** $\lambda, \lambda_1, \lambda_2$. We sweep over the loss coefficients by varying a multiplier $s$ (default $s = 1$) applied to each of them. The impact on pretraining performance is monitored with (a) the large ImageNet dataset (evaluated on 2 tasks), as well as relatively small datasets (b) Flowers-17 and (c) Cars. Performance is not very sensitive to the loss coefficients, especially when they are scaled by $s \in [0.5, 2]$.

Table 9: **Linear classification accuracy (%) on more fine-grained classification datasets**, using ResNet-50 pretrained on ImageNet-100. Both the LooC/LooC++ and AugSelf methods are built on top of the MoCo v2 baseline.

| Method | ImageNet-100 | | iNat-1k | | Cars | SUN397 |
|---|---|---|---|---|---|---|
| | Top-1 | Top-5 | Top-1 | Top-5 | Top-1 | Top-1 |
| MoCo v2 (Chen et al., 2020b) | 81.0 | 95.2 | 36.2 | 62.0 | 33.86 | 46.50 |
| LooC (Xiao et al., 2021) | 79.2 | 94.7 | 44.0 | 69.3 | - | - |
| LooC++ (Xiao et al., 2021) | 81.2 | 95.2 | 46.1 | 71.5 | - | - |
| AugSelf (Lee et al., 2021) | 82.4 | - | - | - | 37.35 | 48.52 |
| MAST-5 | 81.9 | 95.4 | 46.9 | 72.1 | 39.07 | 49.87 |
| MAST-15 | **82.5** | **95.5** | **48.1** | **72.8** | **39.86** | **50.74** |

**More results of fine-grained classification.** As a supplement to Tables 3 and 4, Table 9 provides more results on additional fine-grained classification datasets iNat-1k (Van Horn et al., 2018), Cars (Krause et al., 2013) and SUN397 (Xiao et al., 2010) with relatively large number of classes. The same linear evaluation protocol is adopted, based on the ResNet-50 pretrained on ImageNet-100 for 500 epochs. We compare with both LooC and a new method AugSelf (Lee et al., 2021). AugSelf combines contrastive learning with an auxiliary loss that predicts augmentation parameters to improve SSL generalization. Note both LooC and AugSelf are built on top of the MoCo v2 baseline. It can be observed from the table that our MAST-5 is quite competitive on all datasets, and MAST-15 further improves and consistently achieves the best performance.

# E  DISCUSSION AND COMPARISON WITH MAE-BASED RECONSTRUCTION METHOD

The recent MAE method (He et al., 2022) has achieved great success without explicit learning of augmentation invariances. Hence, our MAST and MAE belong to different categories of SSL methods: MAST falls into the category of discriminative methods that heavily rely on data augmentation, and MAST further improves by disentangled and uncertainty-aware learning of augmentation invariances; while MAE is an autoencoding method that reconstructs the masked out patches of input image. MAE is found to perform decently even without augmentations, partially because the randomness of masking already provides strong regularization on pretraining.

Although the two methods of MAST and MAE will inevitably exhibit different behaviors, we can still get a general sense of how they compare in performance under the same experimental setup. For that, we conduct an apples-to-apples comparison using the same network architecture (ResNet-50). Fast pretraining for 100 epochs is performed on ImageNet. The learned representations are then evaluated by both linear probing and deep fine-tuning protocols, in order to thoroughly compare the discriminative and reconstruction methods. For linear probing, we follow MoCo (He et al., 2020)

Table 10: **MAST vs. MAE on ImageNet.** Top-1 accuracy (%) of ResNet-50 is measured by linear probing and deep fine-tuning after 100 epochs of pretraining.

| Method | Linear | Fine-tuning |
|---|---|---|
| MAE (He et al., 2022) | 37.8 | 77.1 |
| MAST-5 | **70.8** | **78.8** |

to train a linear classifier on frozen features with an initial learning rate of 30 and batch size of 256. For deep fine-tuning, we follow the A3 training recipe in (Wightman et al., 2021) with its specific optimizer and learning rate scheduling.

Table 10 summarizes the comparing results. We can see that, when using ResNet for pretraining, the MAE-based reconstruction method seem unable to learn as strong representations as our MAST-based discriminative method, since MAE yields worse results using both of our evaluation protocols. One future work is to investigate whether our advantage will translate from CNN to Vision Transformer models.

Besides empirical evaluations, we note the main architecture-independent advantages of MAST over MAE are twofold: disentangled augmentation learning and uncertainty learning, which are missing in the latter family of methods to our knowledge. It would be quite interesting to evaluate the two components in the MAE framework and go beyond discriminative learning. Intuitively, if we treat random masking as one type of "augmentation" in MAE, then disentangled learning from *e.g.*, different augmentation strengths (masking ratios) could lead to local and global features learned. On the other hand, uncertainty could be important for ambiguous patch reconstruction in MAE, *e.g.*, when the masking ratio is too high, or when there are barely visual cues around the masked patch location.

