# OpenReview forum: "MAST: Masked Augmentation Subspace Training for Generalizable Self-Supervised Priors"
_ICLR.cc/2023/Conference — ICLR 2023 poster_

### Official Review · Reviewer_oMUf · 2022-10-20

**Confidence:** 4
**Clarity, Quality, Novelty And Reproducibility:** The clarity, quality, novelty, and re…
**Correctness:** 3
**Technical Novelty And Significance:** 3
**Empirical Novelty And Significance:** 3
**Recommendation:** 8

**Strength And Weaknesses:**

Strength:

1. How to learn task-independent representations is a key research problem in SSL.
2. The proposed method shows remarkable improvements in various settings.
3. The paper is well organized and easy to follow.

Weakness:

1. The proposed method is too complex, with 5 hyper-parameters in the loss function. Although authors claim all hyper-parameters do not need to be careful turn, only results on ImageNet-100/1K are provided. How about on other datasets, e.g., CIFAR-10/100?
2. The proposed method shows significant improvements on ImageNet-1K compared with MoCo V2, over 3%, but on ImageNet-100, the improvements drop to below 1%. Why does it happen?
3. The authors do not provide the results of the short-term training, such as 200 epochs. Does the proposed method require long-term training to fully explore the strong augmentations?


**Summary Of The Paper:**

In this paper, the authors propose a method that tend to learn task-independent representations. Specifically, the proposed method uses multiple masks to generate different subspace representations to avoid feature suppression. The proposed method also employs the uncertainty modeling technique against variance from strong augmentations. The experiments demonstrate that the proposed method outperforms SOTA baselines in various downstream tasks.

**Summary Of The Review:**

Although the proposed method is complex and more experiments should be added, it still shows a new way to learn task-independent representations. I recommend this work.

---

> ### Author Response · Authors · 2022-11-14
> **Response to Reviewer oMUf**
>
> Thanks for your recognition of the novelty and quality of our paper. Please find below our response to each of your questions.
>
> **Results with 5 loss hyperparameters on datasets other than ImageNet-100/1K, e.g. CIFAR-10/100.**
>
> As mentioned in Sec. 3.2, our loss function consists of 2 terms borrowed from the VICReg baseline with their default loss coefficients. We introduce 3 more loss terms, and only set their loss coefficients to generate a comparable loss magnitude with that of other terms. We have added Fig. 8 in Appendix D to show our pretraining performance is not very sensitive to the loss coefficients (e.g. when scaled by a factor between 0.5 and 2).
>
> Appendix B (Table 6) also shows the success of MAST on much smaller datasets for pretraining, using the default loss coefficients. Specifically, we pretrain on the small datasets Flowers-17 (1360 images) and Cars (16185 images) respectively, without tuning loss coefficients (only that we lower the pretraining epochs to 100 and use smaller network ResNet-18). Linear probing on each dataset generates strong performance for our MAST-5.
>
> Regarding the suggested CIFAR datasets, we experimented on CIFAR-10 based on MAST-5 using the untuned loss coefficients again. Following the setting in SimSiam, we pretrained the ResNet-18 backbone for 800 epochs for efficiency purposes. Preliminary results of linear evaluation accuracy (\%) seem promising: MAST-5 (92.9), VICReg (92.5), SimSiam (91.8). We leave more experiments (e.g. on CIFAR-100) for future work.
>
> **Why is the improvement larger on ImageNet-1K than on ImageNet-100?**
>
> This is because MAST benefits from longer training as well as larger data. On ImageNet-100, we only pretrain for 500 epochs for fair comparison with other methods that use the same number of epochs. By contrast, we pretrain for 1000 epochs on ImageNet-1K. Clarification is added in the paper.
>
> **Does MAST require long training to fully explore strong augmentations?**
>
> MAST does not require long training to show some benefit, but indeed it does benefit significantly from longer training, which is typically comprised of two stages with increasing number of augmentations (i.e. stronger compositions). Table 8 in Appendix D exemplifies the ImageNet performance as a function of pretraining epochs, from 200 to our default 1000. We observe that 1) MAST-5 outperforms its baseline model VICReg across all epoch numbers, 2) more epochs lead to larger gains, but the gain is still decent (0.7\%) at low epoch number 200.

---

### Official Review · Reviewer_7wWs · 2022-10-21

**Confidence:** 3
**Correctness:** 3
**Technical Novelty And Significance:** 2
**Empirical Novelty And Significance:** 2
**Recommendation:** 6

**Clarity, Quality, Novelty And Reproducibility:**

The paper is generally well written.   A minor comment:

Eq 1: Would using Z,Z' be more appropriate than V,V' since I think the loss is with respect to the embeddings, not views?


The proposed approach was compared with a number of algorithms on a few tasks.

Reproducibility seems ok.


**Strength And Weaknesses:**

Strengths:

1.  Learning embeddings in augmentation subspaces via masking is interesting.

2.  Their results indicate that their approach generally compares favorably.

3.  The paper is generally well written.

Weaknesses:

1.  While performance improves, further discussion/analysis on which subset of the masked embeddings are beneficial to different tasks would be interesting.

**Summary Of The Paper:**

The authors propose learning embedding in augmentation subspaces via masking, which specifies which dimensions are relevant for which augmentation.  In the loss function, the mask is applied to the embeddings before comparison.  Also, the loss function includes a L1 norm term that encourages  masks to be sparse (with fewer features used).   To model uncertainly in the embeddings, the approach learns the mean (replacing the embedding previously) and covariance matrix.  The loss function is normalized by the covariance.   To stablize the learning of the uncertainty model, they use a regularization term to consistency between the Gaussian embeddings before masking via symmetric KL divergence.

They compare with 13 algorithms on ImageNet via linear evaluation and semi-supervised learning.  They also evaluate representations learned on ImageNet for downstream tasks on 5 different datasets via linear classification, object detection, and instance segmentation.  Their results indicate that their approach generally compares favorably.  In their ablation study, they look at the effect of disentangled embedding, scheduling, learned mask, and Gaussian uncertainty.  They analyzed the correlation between masks and add more uncorrelated masks than the more correlated ones.   They found





**Summary Of The Review:**

Learning embeddings in augmentation subspaces via masking is interesting.   Their results indicate that their approach generally compares favorably.

---

> ### Author Response · Authors · 2022-11-14
> **Response to Reviewer 7wWs**
>
> Thanks for the positive feedback and constructive suggestions! Answers to your questions below.
>
> **Which subset of the masked embeddings are beneficial to different tasks?**
>
> This question is inline with our early experiments that aim to find ``what augmentations benefit what tasks''. Fig. 1 is one example. By removing different augmentations from MoCo-based SSL and then evaluating on different tasks, we can quantify the importance of each augmentation for various tasks in a leave-one-out manner. Now with MAST, we are able to quantify the augmentation-task relationships more easily, using the learned feature mask for each augmentation as a bridge. The end task can freely select from the factorized latent space which feature masks or augmentations are relevant to the task. This is much cheaper than the aforementioned leave-one-out strategy.
>
> Specifically, for a classification task and those correctly classified samples, we mask their feature representations to attend to different feature subspaces. Then we simply record the class predictions by differently masked features of each sample (only for the ground-truth class, using the corresponding weights of the classifier). This will help us to examine how each feature subspace, capable of capturing one specific augmentation invariance, responds to or benefits a given classification task. Fig. 6 (Appendix C) exemplifies the class predictions from 5 augmentation-specific subspaces on ImageNet and CUB-200 datasets. We observe a similar trend as in Fig. 1: different tasks benefit from a different set of augmentations. For example, while the augmentations of \{*Color Jitter, Random Grayscale, RandomResizedCrop*\} seem good enough for ImageNet classification, \{*Color Jitter, Random Grayscale*\} are not so useful for CUB-200.
>
> **Eq. 1: Would using Z,Z' be more appropriate than V,V'?**
>
> The baseline loss in Eq. 1 is indeed with respect to the embeddings Z,Z' of views V,V' - we have improved the manuscript to make it clear. But later we extend to model uncertainties of V,V' based on their Gaussian embeddings $\mathcal{N}(\mu, \Sigma)$ in Eqs. 4-7. Hence we stick to the same loss formulation using V,V' throughout the paper for simplicity of notation, but with clear descriptions about the associated embeddings.

---

> > ### Comment · Reviewer_7wWs · 2022-11-15
> > **comment on author response**
> >
> > "Optimal augmentations (subspaces) are task-dependent" in Appendix C is related to my question/comment.  I think it's important and would include it in the paper.  Since MAST can disentangle the augmentations in the embedding space, how to tell which subset of augmentations (masked emeddings) is more important for a downstream task would strengthen your claim for disentanglement.   Fig. 6 in Appendix C helps answer the question.    The masks are on the embeddings (Z), but the representation used in downstream tasks is in y (before projecting into Z).   How do you find the corresponding representation subspace in y for a (masked) embedding subspace in Z?
> >
> > I would use Z,Z' or V,V' when they are actually used in a loss function.  The "overall/combined" loss function could have both Z,Z' and V,V'.  For example: L(Z,Z',V,V') = L1(Z,Z',..) + L2(V,V',...) + ...    I think this would clarify which layer in the network that a loss component is focusing on.

---

> > > ### Author Response · Authors · 2022-11-16
> > > **Re: comment on author response**
> > >
> > > Thanks for your feedback! We have improved the manuscript according to the comments on the equation notation and Fig. 6 (which is referenced under Eq. 3 to save space). For clarification, in Fig. 6, the subspaces at the representation level (y) are obtained using the embedding masks that are simply downsampled. We observed similar invariance properties (in terms of invariance metric) for the masked representations and masked embeddings. This reaffirms our claim in the paper that the disentanglement at the embedding level will have a disentanglement effect at the representation level.

---

> > > > ### Comment · Reviewer_7wWs · 2022-11-17
> > > > **comment on author response**
> > > >
> > > >  > the subspaces at the representation level (y) are obtained using the embedding masks that are simply downsampled
> > > >
> > > > Please further elaborate on "downsampling."  (Also, Z could have a smaller number of dimensions than y, maybe not in your case).

---

> > > > > ### Author Response · Authors · 2022-11-18
> > > > > **Re: comment on author response**
> > > > >
> > > > > In our case, the embedding Z has higher dimensionality (4096) than representation y (2048). Hence we downsample Z's mask by a factor of 2 using Bilinear interpolation, in order to perform masking for y. Upsampling is needed when Z is lower dimensional than y. Note such mask resampling is only for visualization purposes in Fig. 6, where we train a classifier on top of frozen y, and then mask y using the resampled mask of Z to study the corresponding class prediction. Alternatively, we have conducted such analysis based on the classifier and mask both applied to Z (so no mask resampling is required, but this should be avoided since eventually we need to drop all the embeddings Z), In this case, we observe similar distribution of the class predictions across masked subspaces.

---

> > > > > > ### Comment · Reviewer_7wWs · 2022-11-18
> > > > > > **comment on author response**
> > > > > >
> > > > > > Z = g(y), where g(.) is the projector.   Mask M is also learned, not sure if M is also a function of y or not (seems to be not in the paper).
> > > > > >
> > > > > > In your downsampling of a mask for Z, projector g seems to be not considered.  Mask m has the same "ordering" of dimensions as Z. However, I don't think the "ordering" of dimensions in Z has any relationship with the "ordering" of dimensions in y.   Consequently, it seems finding the corresponding mask for y needs to consider projector g.    Can you elaborate on why finding the corresponding mask for y doesn't need to consider projector g?    (If M is a function of y: M = h(y), does h need to be considered as well?)

---

> > > > > > > ### Author Response · Authors · 2022-11-19
> > > > > > > **Re: comment on author response**
> > > > > > >
> > > > > > > That is a great question, and is one of our on-going investigations of mask transferability. We note that mask M is trained on embeddings Z, and M should also be an implicit function of representations y because of the projector g that connects y and Z. However, it is true that y and Z are not perfectly aligned dimension-wise. Hence it is far from ideal to mask y using the downsampled mask of Z, and we stress that the results in Fig. 6 are entirely empirical. There are two observations: 1) there is a decent cosine similarity between y and downsampled Z, suggesting some level of alignment between the two feature spaces; 2) the masks in Fig. 3(a) demonstrate clustering patterns along neighboring dimensions, which offers some flexibility to handle the misalignment issue. These observations are part of the reasons why we obtained similar distributions of class predictions, based on masked y using the downsampled masks of Z vs. masked Z using the original masks, despite the misalignment between y and Z.
> > > > > > >
> > > > > > > That said, we decided to improve our analysis in a rigorous setting. First, we updated Fig. 6 in the revised manuscript - the class predictions are now studied on masked Z (so no mask downsampling is required), and we have updated the text to reflect this. The new Fig. 6 still points to the same conclusion: different tasks benefit from different augmentations (masked embedding subspaces). Second, we plan to learn two sets of masks M_y and M_Z for y and Z respectively, using the same loss during pretraining. This enables careful studies of the relationship between M_y and M_Z, and generalization of M_y (and its induced subspaces of the desired representations y).

---

> > > > > > > > ### Comment · Reviewer_7wWs · 2022-11-24
> > > > > > > > **comment on author response**
> > > > > > > >
> > > > > > > > > the class predictions are now studied on masked Z
> > > > > > > >
> > > > > > > > Your proposed method for downstream tasks uses features from y not features from Z, so analyzing predictions based on masked features from Z does not seem to be relevant.
> > > > > > > >
> > > > > > > > What are the differences/tradeoffs between learning M_y and M_Z?

---

> > > > > > > > > ### Author Response · Authors · 2022-11-28
> > > > > > > > > **Re: comment on author response**
> > > > > > > > >
> > > > > > > > > We humbly argue that the analyses in appendix are important because they validate the success of our disentangled invariance learning at targeted Z level --- Fig. 7 shows augmentation invariances are indeed captured in each embedding subspace, while the class prediction analysis in Fig. 6 shows the embedding subspaces are responsive for different tasks, as we have expected.
> > > > > > > > >
> > > > > > > > > It will be our future work to analyze class predictions at the y level. Examining the properties of y can help us to understand how and why representations generalize across various tasks as shown in many of our experiments. In practice, representations do not need masking during transfer learning, but class prediction using masked y does (only for analysis purpose). In our previous attempt, y is simply masked by a downsampled version of Z's mask, and we find empirical evidence that differently masked y also benefits different classification tasks - this suggests some disentanglement effect at the representation level too. Another interesting direction is to learn masks M_y jointly with M_Z (at different feature layers) and then only use M_y for disentanglement studies. This may enable more rigorous analysis without the misalignment issue between y and Z, but the joint mask learning would be nothing about pushing performance limits of MAST.

---

### Official Review · Reviewer_JyME · 2022-10-22

**Confidence:** 4
**Correctness:** 2
**Technical Novelty And Significance:** 2
**Empirical Novelty And Significance:** 3
**Recommendation:** 6

**Clarity, Quality, Novelty And Reproducibility:**

The detailed comments are described in the previous section. In summary,
- Clarity :: The method description is unclear.
- Quality :: The empirical results seem strong, but some important comparisons are missing.
- Novelty :: I think the proposed idea is somewhat novel.
- Reproducibility :: All the hyperparameters are well-described.


**Strength And Weaknesses:**

Strengths
- This paper aims at learning more generalizable representations, which is an important problem in the SSL literature.
- The proposed method outperforms previous SSL methods under various downstream tasks.

Weaknesses
- Method explanation is unclear: how can Eq (2) learn augmentation-specific masks $m_i$? For example, how to know $m_1$ is related to color jittering, and $m_2$ is related to Gaussian blur, as shown in Figure 3a? Eq (2) does not force any relationship between a mask $m_i$ and an augmentation operation $o_j$.
  - I'm wondering whether MAST uses $K$ forward passes for each training iteration as LooC [1] did. If not, how to learn augmentation-specific embedding spaces? This could be critical because Eq (2) may cause misunderstanding.
- Recent MAE-based approaches do not learn augmentation-invariant representations. What is the advantage of this paper over them? This paper should be compared with the MAE methods.
- The hyperparameter choices of λ = 25d/K, λ1 = 600/(dK) and λ2 = 25 seem to be carefully chosen. So, I think ``work well for all experiments without the need of careful tuning.'' is not true. If the authors want to claim that the sentence is true, they should conduct experiments with varying λ, λ1, and λ2 and also should verify that performance is not sensitive to the hyperparameter choices.
- There exist many fine-grained classification tasks, but Table 4 only provides two benchmarks, CUB and Flowers. More experiments with the tasks should be provided (see SimCLR [2], BYOL [3], or AugSelf [4] papers).
  - This paper should be compared with [4]: [4] achieves 45% top1 accuracy on CUB-200 and 88% on Flowers-102.
- There is no analysis of the correlation between each augmentation-specific embedding space and augmentation-related information (e.g., see Table 1 & Figure 3 in LooC [1] or see Table 8 & Figure 5 in AugSelf [4]).
- There is no analysis of uncertainty. Such an accuracy comparison in Table 5 does not explain how the uncertainty affects learning representations. Also, this paper should provide some empirical evidence verifying that the uncertainty is successfully learned.

[1] Xiao et al., What should not be contrastive in contrastive learning, ICLR 2021 \
[2] Chen et al., A Simple Framework for Contrastive Learning of Visual Representations, ICML 2020 \
[3] Griall et al., Bootstrap Your Own Latent: A New Approach to Self-supervised Learning, 2020 \
[4] Lee et al., Improving Transferability of Representations via Augmentation-Aware Self-Supervision, NeurIPS 2021


**Summary Of The Paper:**

This paper proposes a self-supervised learning method, Masked Augmentation Subspace Training (MAST), which learns more generalizable representations by modeling augmentation-specific invariance properties. This paper shows that MASK outperforms existing SSL methods under various downstream tasks, including semi-supervised learning, detection/segmentation, and fine-grained classification.


**Summary Of The Review:**

Although this paper successfully achieves strong empirical results in various downstream tasks, the method descriptions are unclear, and some important comparisons and analyses are missing. I think they are critical, so I vote for rejection.

---

> ### Author Response · Authors · 2022-11-15
> **Response to Reviewer JyME (1/2)**
>
> Thank you for the constructive feedback and helpful suggestions for our manuscript. We respond to specific comments below.
>
> **How can Eq. (2) learn augmentation-specific masks? Need K forward passes?**
>
> Please refer to our training details in Sec. 3.2, where we adopt a two-stage pretraining mechanism. In stage 1, one augmentation is sampled at a time to ensure good training of the corresponding subspace and mask. In stage 2, there are K augmentations for each input, and we only need to mask the feature embeddings K times to compute all subspace losses and then update the embeddings and K masks jointly (so no K forward passes). Such joint training loses the mask-augmentation association, but the goal is to encourage collaborative mask learning (and mask dimension sharing) from stronger augmentation compositions. Fig. 3(a) shows that meaningfully separable subspace masks are still learned after stage 2, and Fig. 7 (in Appendix C) confirms that each subspace is still specialized for the corresponding augmentation invariance. Clarification is added under Eq. (2).
>
> **What is the advantage of MAST over MAE-based approaches?**
>
> MAST and MAE belong to different categories of SSL methods. MAST falls into the category of discriminative methods that heavily rely on data augmentation, and MAST further improves by disentangled and uncertainty-aware learning of augmentation invariances. MAE is an autoencoding method that reconstructs the masked out patches of input image. MAE is found to perform decently even without augmentations, partially because the randomness of masking already provides strong regularization on pretraining. These two methods exhibit different behaviors, but lack apples to apples comparisons with the same network architecture -- MAE and follow-up methods are implemented by ViT, while discriminative methods are mostly ResNet-based (including our MAST).
>
> Nevertheless, we can still roughly compare the two methods in the ResNet world, using an MAE-type method with ResNet implementation. Context Encoder [CVPR'16], with the same ResNet-50 backbone, inpaints random masked image regions (similarly to MAE). The PCL paper listed in our Table 1 reports more comparison results in terms of ImageNet linear classification Top-1 accuracy: PCL= Context Encoder+SimCLR (71.0\%) vs. Context Encoder (43.7\%) vs. SimCLR (69.3\%). Note our MAST achieves 74.9\%. Hence in this ResNet setting, MAE-style methods alone seem not enough to learn as strong representations as discriminative methods.
>
> It is an interesting direction to investigate whether the advantage of our ResNet-based MAST over MAE/Context Encoder will translate to ViT models, given that MAE is shown to significantly benefit from transformer-specific architectural designs. We leave for future work the MAST-MAE comparison purely based on ViT. The ideas behind MAST can also shed light on how to further improve MAE, e.g. via uncertainty modeling for reconstruction, and masking augmentation learning to robustify reconstruction.
>
>
> **Sensitivity analysis of loss coefficients $\lambda,\lambda_1,\lambda_2$.**
>
> Our loss function consists of 2 terms borrowed from the VICReg baseline with their default loss coefficients. We introduce 3 more loss terms, with their loss coefficients $\lambda,\lambda_1,\lambda_2$ only set to generate a comparable loss magnitude with that of other terms. We have added Fig. 8 in Appendix D to show our pretraining performance on different datasets is not very sensitive to the loss coefficients (e.g. when scaled by a factor between 0.5 and 2).
>
> **More fine-grained classification results and compare with AugSelf (on CUB-200 and Flowers-102).**
>
> Due to time constraints, we only evaluated fine-grained classification on 3 more datasets---iNat-1k, Cars and SUN397 with relatively large number of classes (compared to CUB-200 and Flowers-102). We compare our approach with LooC and AugSelf, both of which are built on top of the MoCo v2 baseline. Note we do not compare with AugSelf on CUB-200 and Flowers-102 because its results are not directly comparable on these two datasets (the authors claim in the footnote that they reproduced the AugSelf's baseline MoCo v2, whose results are much higher or lower than the official numbers). Table 9 (Appendix D) summarizes the results on the 3 new datasets where our MAST outperforms both LooC and AugSelf consistently.

---

> > ### Author Response · Authors · 2022-11-15
> > **Response to Reviewer JyME (2/2)**
> >
> > **Analysis of the augmentation-related information in each feature subspace.**
> >
> > The suggested papers (LooC and AugSelf) both aim to preserve augmentation invariance as well as augmentation-specific information (e.g. location- or color-sensitive info). Their analysis is mostly focused on the latter goal, by evaluating how well the augmentation characteristics are preserved in the additional feature heads that are trained to be augmentation-sensitive. The evaluations include transfer learning on particular tasks like position-sensitive object localization, as well as solving augmentation-aware pretext tasks. Since our MAST method focuses on learning disentangled augmentation invariances rather than augmentation sensitivities, we focus our analysis on the invariance property.
> >
> > Specifically, we measure how well each of our masked feature subspaces is invariant to the corresponding augmentation. Fig. 7 in Appendix C shows the invariance metric on ImageNet when we vary the augmentation parameters of 3 standard augmentations -- Color Jitter, Gaussian Blur and RandomResizedCrop. We exclude other 2 (Random Flip, Random Grayscale) whose augmentation parameters are discrete. The observation is that our masked feature subspaces show great robustness under various augmentation intensities, especially when compared to the VICReg baseline that learns all augmentation invariances in an unfactorized way.
> >
> > One might ask: will we loose all augmentation-sensitive information when we learn factorized invariances from those augmentations? Short answer is no, as evidenced by our strong performance on various downstream tasks that rely on specific augmentation characteristics (e.g. color-sensitive bird classification on CUB-200). We hypothesize that different feature subspaces may offer complementary information about augmentations for different tasks to choose from. For example, in Fig. 3(b), the color information suppressed in the subspace invariant to Random Grayscale is available in the subspace invariant to RandomResizedCrop.
> >
> > To further support the above hypothesis, we follow the AugSelf paper to classify the rotation degrees on STL-10 dataset, using representations pretrained without the Rotate augmentation. Table 7 in Appendix C shows competitive performance of MAST-5 (63.47\%) vs. AugSelf (64.61\%) which is explicitly trained to be augmentation-predictive. This validates that we can learn invariances to standard augmentations while maintaining sensitivity to the unseen rotation. In other words, MAST is able to preserve augmentation-invariant and -specific info implicitly in separate subspaces.
> >
> > **Missing analysis of uncertainty.**
> >
> > As mentioned in the last paragraph of the experiment section, Appendix B visualizes example uncertainty estimations that are meaningful (Fig. 5(a)), and explains how high uncertainty downweights the distance between ambiguous views to avoid similarity mismatch, and how uncertainty reflects learning difficulty (Fig. 5(b)). Comparison with a related uncertainty model is also provided in Table 6.

---

> > > ### Comment · Reviewer_JyME · 2022-12-01
> > > **Response to Authors**
> > >
> > > Thank you for your time and efforts in your response. Some of my concerns are resolved, but others are still remaining. So, I have increased my score from 3 (Reject) to 5 (Weak Reject). If all the concerns are fully resolved, I'm willing to increase my score further (after discussion with other reviewers).
> > >
> > > ---
> > >
> > > **How can Eq. (2) learn augmentation-specific masks? Need K forward passes?**
> > >
> > > Thank you for your explanation. I now understand how they are associated. IMHO, the authors should discuss the two-stage training details in the Method section because it is still unclear how to associate each subspace with each augmentation. In addition, the terminology of "augmentation" is also unclear: $K$ augmentations = a single augmented view with $K$ atomic operations or $K$ different augmented views? I think authors should clarify this.
> > >
> > > ---
> > >
> > > **What is the advantage of MAST over MAE-based approaches?**
> > >
> > > Conceptually, MAST and MAE can be applied into both Conv and ViT architectures (e.g., MAE can be applied into ConvNet architectures [1]). So I think the architectural difference is not important. I agree that they fall into different categories, but that cannot be the advantage of MAST over MAEs. I'm still wondering what is the (conceptual) advantage of the MAST over MAEs.
> > >
> > > Also, under linear evaluation, discriminative methods (e.g., MAST) could be better than reconstruction methods (e.g., MAE), but it could be not under fine-tuning setup. Therefore, in my opinion, considering only linear evaluation is not enough to compare them comprehensively.
> > >
> > > ---
> > >
> > > **Sensitivity analysis of loss coefficients**
> > >
> > > Thank you for the additional results.
> > >
> > > ---
> > >
> > > **More fine-grained classification results and compare with AugSelf (on CUB-200 and Flowers-102).**
> > >
> > > First, thank you for your experiments in the three fine-grained benchmarks. However, I cannot understand why the authors do not compare MAST with AugSelf on the CUB and Flowers benchmarks. AugSelf results are also obtained from the same pretraining dataset and the same architecture. Why the results are not fairly comparable?
> > >
> > > ---
> > >
> > > **Analysis of the augmentation-related information in each feature subspace**
> > >
> > > Thank you for the detailed explanation. I have few more questions: the subspace of color jittering is less invariant to random cropping augmentations? Also, are some subspaces bad to solve rotation prediction tasks? (e.g., a subspace invariant to random cropping could be not useful for the task) These experiments could show that the subspaces are well-disentangled.
> > >
> > > ---
> > >
> > > **Missing analysis of uncertainty**.
> > >
> > > Thank you for the explanation. I have checked the Appendix B and that is enough to resolve this concern.
> > >
> > > [1] Li et al., Architecture-Agnostic Masked Image Modeling -- From ViT back to CNN, 2022

---

> > > > ### Author Response · Authors · 2022-12-05
> > > > **Response to Reviewer Feedback**
> > > >
> > > > Thanks for your feedback! Below we respond to the follow-up questions to hopefully address your remaining concerns.
> > > >
> > > > > The authors should discuss the two-stage training details in the Method section. In addition, the terminology of "augmentation" is also unclear: K augmentations = a single augmented view with K atomic operations or K different augmented views?
> > > >
> > > > Thanks for the advise. In the next version of this manuscript, we will include more details about the two-stage training after Eq.(2), and clarify that 'K augmentations' means K augmentation operators applied on a single view.
> > > >
> > > > > MAST and MAE can be applied into both Conv and ViT architectures. What is the conceptual advantage of MAST over MAE? Also, considering only linear evaluation is not enough to compare them comprehensively.
> > > >
> > > > We totally agree that the MAST-MAE comparison should be nothing about any network architectural difference, and we do not claim our advantage comes from that. What prevented us from an apples-to-apples comparison between MAST and MAE was only the lack of papers that do so using the same architecture (ConvNet or ViT) and experimental setup, as well as the lack of time for us to re-implement both methods under the same setting during the previous paper discussion stage.
> > > >
> > > > After that, we have actually found and followed the suggested A2MIM paper to compare MAST and MAE under the same fast pretraining setting --- pretraining ResNet-50 for 100 epochs, and then evaluating it by both linear probing (Lin) and RSB A3 fine-tuning (FT) on ImageNet. The Top-1 accuracies (\%) from Lin/FT are: 37.8/77.1 for MAE (reported) vs. 70.8/78.8 for our MAST-5. This reaffirms our advantage over the MAE-based reconstruction method, at least in the ConvNet world. We will add the results in Appendix, and plan to implement ViT-based MAST approach to compare with MAE-type ViT methods in our future work.
> > > >
> > > > Besides empirical evaluations, we note the main architecture-independent advantages of MAST over MAE are twofold: disentangled augmentation learning and uncertainty learning, which are missing in the latter family of methods to our knowledge. It would be quite interesting to evaluate the two components in the MAE framework (thus going beyond our discriminative framework). Intuitively, disentangled learning from e.g. different ratios of random masking (the only "augmentation" type in MAE) could lead to local and global features simultaneously learned. Uncertainty could be important for ambiguous patch reconstruction, e.g. when the masking ratio is too high, or when there are barely visual cues around the masked patch location. Such discussions will be included in our paper to shed light on future directions.
> > > >
> > > > > Why the authors do not compare MAST with AugSelf on the CUB and Flowers benchmarks?
> > > >
> > > > As mentioned before, this is because AugSelf has re-implemented its baseline methods that are not quite comparable. Specifically, when AugSelf is combined with the MoCo baseline (see Table 3 of the AugSelf paper), the authors footnote that they have reproduced the MoCo baseline whose results are drastically different from the reported numbers for MoCo (2nd row vs. 6th row in Table 3) on CUB-200 and Flowers-102. Hence we exclude MoCo + AugSelf from our comparisons for the aforementioned datasets. Same story for SimSiam + AugSelf. The SimSiam baseline obtains classification accuracies (\%) of 38.4/83.6/85.9 for CUB-200/Flowers 5-shot/Flowers 10-shot. Such results are already much higher than those of SOTA methods listed in Table 3, and also different from our implementation results for SimSiam (37.3/69.5/80.3). As a result, we do not consider CUB-200 and Flowers-102 datasets for AugSelf comparison purposes.
> > > >
> > > > > Is the subspace of color jittering less invariant to random cropping augmentations? Are some subspaces bad to solve rotation prediction task?
> > > >
> > > > Great questions! To address the questions, we have first extended from Fig. 7 (Appendix C) to study the invariance metric across 3 subspaces on ImageNet. When the *Color Jitter* subspace is tested against the *RandomResizedCrop* augmentation (with cropping size ratio 0.5:0.1:0.9), the invariance metric 0.785/0.887/0.945/0.973/0.982 is apparently lower than that of the *RandomResizedCrop* subspace (0.961/0.978/0.989/0.994/0.996). This demonstrates the specialization of different subspaces (to corresponding augmentations) and their disentanglement. More results will be added to the Appendix. For rotation prediction on STL-10, we have similar observations. The best rotation-predictive subspace (out of 5 augmentation subspaces) is the *Color Jitter* one, obtaining 51.22\% accuracy for rotation classification. The *RandomResizedCrop* subspace is less useful (39.54\%), while the *Random Flip* subspace performs worst (11.27\%). This shows our subspaces are not only well-disentangled but also collaborative in the prediction of unseen rotation.

---

> > > > > ### Comment · Reviewer_JyME · 2022-12-05
> > > > > **Response to Authors**
> > > > >
> > > > > Thank you for the detailed response.
> > > > >
> > > > > I'm still concerned about the comparison with AugSelf. I already checked the footnote of AugSelf and I cannot agree with your opinion. I'm wondering why such re-implemented results are not comparable? Your code or LooC is not "re-implemented"? All experiments are conducted by a kind of re-implementation. So I think it cannot be the reason for no comparison. If the experimental setup is the same, you should compare/report them. I also find that AugSelf's code is publicly available, so I think the AugSelf results are reproducible. In contrast, I cannot find the LooC code (as noted by the authors of AugSelf), so I'm not sure that which result (AugSelf vs LooC) is more convincing. Hence, I think you should compare your results with both rather than only the worse one. Honestly, I'm very unsatisfactory in this response.
> > > > >
> > > > > Other responses seem reasonable, and I hope your manuscript will be updated based on the responses.
> > > > >
> > > > > If I'm missing something, please let me know.

---

> > > > > > ### Author Response · Authors · 2022-12-05
> > > > > > **Response to Reviewer Feedback**
> > > > > >
> > > > > > Thanks for your prompt reply! Also glad to see most of your concerns have been addressed - we will definitely update the manuscript accordingly.
> > > > > >
> > > > > > Regarding the comparison with AugSelf, we do realize that adding it to Table 4 will lead to a more comprehensive comparison under the same experimental setup. Hence, as suggested, we will make sure to include the results of both MoCo + AugSelf and SimSiam + AugSelf. This way, Table 4 will contain results of LooC and two versions of AugSelf, which are all directly cited from the respective papers to avoid issues like code unavailability or differences from our re-implementation. For such comparisons, 1) we will make clear that the AugSelf authors have re-implemented the MoCo/SimSiam baselines whose results can be different from the official baseline numbers. 2) for completeness, we will further report the accuracy gains of MAST over its baseline VICReg, as similarly done by AugSelf over its different baselines. This could be a fairer comparison to evaluate the core contributions of each method. The accuracy gains (%) for CUB-200/Flowers 5-shot/Flowers 10-shot are: 2.8/1.8/3.7 from MAST-5 vs. 3.6/4.1/4.4 from MAST-15 vs. 4.8/3.2/3.3 from AugSelf over MoCo vs. 6.9/2.8/2.4 from AugSelf over SimSiam. We can see that our MAST approach remains competitive especially on the Flowers dataset.
> > > > > >
> > > > > > Please let us know if you have more questions. Thanks!

---

> > > > > > > ### Comment · Reviewer_JyME · 2022-12-08
> > > > > > > **Response to Authors**
> > > > > > >
> > > > > > > Thank you for your efforts in this response! Since all my concerns have been addressed, I have increased my score.

---

> > > > > > > > ### Author Response · Authors · 2022-12-08
> > > > > > > > **Response to Reviewer Feedback**
> > > > > > > >
> > > > > > > > That's great news and thanks for your constructive comments! We will make sure to integrate them into the final paper.

---

### Official Review · Reviewer_Psus · 2022-11-02

**Confidence:** 5
**Clarity, Quality, Novelty And Reproducibility:** Their paper is in good clarity and qu…
**Correctness:** 4
**Technical Novelty And Significance:** 3
**Empirical Novelty And Significance:** 3
**Recommendation:** 6

**Strength And Weaknesses:**

Strength:
- Their paper has pointed out the problem of current deep learning field is that current model or data augmentation lacks of generalization.

- Weakness:

In their data augmentation part of 3.2, they set K=5 and K=15. Is there any performance trend of the different number of K?


**Summary Of The Paper:**

This paper illustrates method (MAST) on how to transfer data augmentation to a downstream task for Self-Supervised Learning (SSL) methods.

From their paper, their conclusion is that their method can show important role of data augmentation in creating useful invariance priors during SSL.

Here are their contribution:
- introduce MAST to make SSL representations disentangled and uncertainty-aware to
effectively encode different augmentation invariances for good generalization.
•show MAST is efficient, is resistant to feature suppression, and achieves state-of-the-art
downstream performance on diverse vision tasks without presuming any task information
during pre-training.
• provide interesting insights about how different augmentations are related in SSL and
how uncertainty reflects learning difficulty and impacts learning adaptively.


**Summary Of The Review:**

This paper mainly contributes a MAST by proposing a unified framework to embed all the distinct priors of invariance into
feature representations that can be readily used for different tasks. Generally, it is trying to solve a problem of generalization and downstream task. Generally, their method has shown a good direction on self supervised learning.

---

> ### Author Response · Authors · 2022-11-14
> **Response to Reviewer Psus**
>
> Thank you for the positive feedback on our work. Regarding the question of **performance trend over K**, please refer to Fig. 4(b) where K increases from 1 to 19. We can observe from the figure that 1) performance generally increases with K, due to the capability of MAST to learn from various augmentations in a factorized way, increasing the richness of the latent representation that could otherwise potentially be harmed by contradicting augmentations. 2) Augmentation diversity facilitates generalization: the first few augmentations are selected to be distinct from each other and improve downstream performance quickly due to the diversity of the latent space, while performance plateaus when K is around 15 since there is more redundancy in the larger space of augmentation types.

---

### Author Response · Authors · 2022-11-17
**Reviewer comments incorporated in revised paper**

Thanks to all reviewers for the positive feedback and helpful suggestions! We’ve uploaded a new version of the paper that incorporates most of the reviewer comments. Beside more clarifications as suggested by different reviewers, other changes include:
- **[JyME]** mentioning the two-staged training details in Sec 3.2 to elaborate on the learning mechanism of augmentation-specific masks
- **[JyME, oMUf]** adding sensitivity analysis of loss coefficients in Fig. 8 and Appendix D
- **[JyME]** adding more fine-grained classification results and comparisons with AugSelf in Table 9 (Appendix D)
- **[JyME]** adding Fig. 7 and Appendix C to quantify the augmentation invariance learned for each subspace.
- **[JyME]** adding Table 7 and Appendix C to help understand the preservation of augmentation-related information in different subspaces
- **[7wWs]** adding and mentioning Fig. 6 in Appendix C to examine which augmentation subspaces benefit what tasks
- **[7wWs]** fixing equation notations as appropriate
- **[oMUf]** adding Table 8 in Appendix D to study the impact of pretraining epochs

Please let us know if you have any further questions, and we are happy to follow up! Thanks for your time!

---

### Decision · Program_Chairs · 2023-01-20

**Decision:**

Accept: poster

**Justification For Why Not Higher Score:**

Based on all the reviews, this paper is a solid accept but the contribution and novelty has not met the bar for spotlight.

**Justification For Why Not Lower Score:**

All reviewers agree to accept the paper due to its technical contributions.

**Metareview: Summary, Strengths And Weaknesses:**

The reviewers agree that the paper is to mitigate an important research problem that current model or data augmentation lacks of generalization. The idea of learning embeddings in augmentation subspaces via masking is interesting and its effectiveness has also been validated by empirical results.

**Note From Pc:**

if the above contains the word "oral" or "spotlight" please see: "oral" presentation means -> notable-top-5% and "spotlight" means -> notable-top-25%. As stated in our emails, we are disassociating presentation type from AC recommendations